# The Optimal Cutoff Value of Tumor Markers for Prognosis Prediction in Ampullary Cancer

**DOI:** 10.3390/cancers15082281

**Published:** 2023-04-13

**Authors:** Seungho Lee, Hongbeom Kim, Heeju Sohn, Mirang Lee, Hyesol Jung, Youngjae Jo, Youngmin Han, Wooil Kwon, Jin-Young Jang

**Affiliations:** 1Departments of Surgery, Cancer Research Institute, Seoul National University College of Medicine, Seoul 03080, Republic of Korea; 2Division of Hepatobiliary and Pancreatic Surgery, Department of Surgery, Samsung Medical Center, Sungkyunkwan University School of Medicine, Seoul 06351, Republic of Korea

**Keywords:** ampullary cancer, tumor marker, cutoff value, prognosis, overall survival

## Abstract

**Simple Summary:**

This study aimed to determine the relationship between the prognosis of ampullary cancer (AC) and the level of the tumor marker CA 19-9 (carbohydrate antigen 19-9), and to identify the optimal cutoff values for CA 19-9. The study enrolled 385 patients who underwent curative resection for AC. The C-tree method was used to determine the optimal cutoff value for CA 19-9, which was found to be 46 U/mL. This new cutoff value was a statistically significant prognostic factor for AC and may be used to evaluate the prognosis of AC and determine treatment strategies such as surgical treatments and adjuvant chemotherapy.

**Abstract:**

Background: Carbohydrate antigen 19-9 (CA 19-9) is a representative tumor marker used for the diagnosis of pancreatic and biliary tract cancers. There are few published research results that can be applied to actual clinical practice for ampullary cancer (AC) alone. This study aimed to demonstrate the relationship between the prognosis of AC and the level of CA 19-9, and to determine the optimal thresholds. Methods: Patients who underwent curative resection (pancreaticoduodenectomy (PD) or pylorus preserving pancreaticoduodenectomy (PPPD)) for AC at the Seoul National University Hospital between January 2000 and December 2017 were enrolled. To determine the optimal cutoff values that could clearly stratify the survival outcome, the conditional inference tree (C-tree) method was used. After obtaining the optimal cutoff values, they were compared to the upper normal clinical limit of 36 U/mL for CA 19-9. Results In total, 385 patients were enrolled in this study. The median value of the tumor marker CA 19-9 was 18.6 U/mL. Using the C-tree method, 46 U/mL was determined to be the optimal cutoff value for CA 19-9. Histological differentiation, N stage, and adjuvant chemotherapy were significant predictors. CA 19-9 36 U/mL had marginal significance as a prognostic factor. In contrast, the new cutoff value, CA 19-9 46 U/mL, was found to be a statistically significant prognostic factor (HR: 1.37, *p* = 0.048). Conclusions: The new cutoff value of CA 19-9 46 U/mL may be used for evaluating the prognosis of AC. Therefore, it may be an effective indicator for determining treatment strategies such as surgical treatments and adjuvant chemotherapy.

## 1. Introduction

Ampullary cancer (AC) has a better prognosis than other periampullary cancers [1]. However, compared to other gastrointestinal cancers, AC is a cancer that still has a high recurrence rate [2]. Surgical resection is an essential therapeutic option for AC. A pancreaticoduodenectomy (PD) is considered the optimal treatment for AC; however, this procedure is complex and is associated with severe complications and high morbidity. Therefore, in patients in whom aggressive surgery is difficult, endoscopic papillectomy or ampullectomy may be an alternative treatment [3]. Additionally, the diagnostic accuracy of a preoperative biopsy of AC is 67.3–81.0%, and an accurate evaluation of the T stage is also difficult. As a result, some cases of excessive surgery have been reported. It is difficult to determine the optimal treatment strategy, and thus prognostic factors need to be identified before surgery. Tumor markers are molecules produced by cancerous or other normal tissues in response to tumors. They are detected in the blood, body fluids, and tissues. Tumor markers may be used in cancer screening, diagnosis, prognosis assessment, treatment response prediction, and recurrence monitoring [4,5]. The carbohydrate antigen 19-9 (CA 19-9), a representative tumor marker, is used widely in the diagnosis of a pancreatobiliary malignancy, and many studies have been conducted on its predictive value [6,7]. Although currently the cutoff value that is used in clinical practice is used regardless of the carcinomas, the disease profiles of pancreatic cancer, biliary tract cancer, gallbladder cancer, and AC are different, and a study suitable for each type of cancer is needed. However, most studies thus far have focused on pancreatic cancer [8,9], and studies on AC are scarce. While there have been studies that included AC, they were conducted in other biliary tract cancers [10]. Therefore, the purpose of this study was to determine the usefulness of tumor markers as prognostic factors for AC alone and to measure the cutoff values of these tumor markers.

## 2. Materials and Methods

### 2.1. Patients

Patients who underwent a curative resection (PD or PPPD) for AC at the Seoul National University Hospital between January 2000 and December 2017 were included in this study. A total of 462 patients with AC who underwent PD and PPPD between 2000 and 2017 at Seoul National University Hospital were identified. Patients with the following criteria were ruled out. A total of 21 patients who underwent surgeries other than PD or PPPD were excluded. Five patients who received an R2 resection, one patient with distant metastasis, forty-one patients who had insufficient tumor markers, and nine patients with a histology other than adenocarcinoma and mucinous carcinoma were also excluded. Finally, the data of 385 participants were analyzed (Figure 1).

### 2.2. Data Collection

This retrospective study was based on prospectively collected data. In this study, data including the age, sex, BMI, ASA score, presence of DM, total bilirubin level, preoperative CEA and CA19-9 levels, type of operation, operation time, and amount of expected blood loss were analyzed. CA 19-9 values and total bilirubin values were all taken immediately before surgery, regardless of whether biliary drainage was performed or not. For the postoperative data, histologic differentiation, TNM stage, pathological type of tumor, duration of hospital stay, complications, adjuvant treatment, and recurrence were analyzed. (The 7th edition of the AJCC cancer staging was used.) Adjuvant chemotherapy was mainly performed in cases with regional lymph node metastasis. In the past, 5-FU was the main adjuvant chemotherapy, but from around 2012, gemcitabine and cisplatin were adopted as the main therapy. The dosage of 5-FU was mainly 400 mg/m^2^, and folinic acid or cisplatin was additionally used. The doses of gemcitabine and cisplatin varied from case to case, but were approximately 1000 mg/m^2^ and 25 mg/m^2^. Adjuvant radiation therapy was recommended when margin positive or lymph node metastasis was severe, but the final decision was made by the Department of Radiation Oncology. The median total amount of radiation per dose was about 50 Gy (range; 45–54 Gy). The Institutional Review Board of Seoul National University Hospital approved the collection of the data, storage, and analysis (SNUH 2206-214-1336).

### 2.3. Statistical Analysis

The categorical variables were expressed as numbers and proportions, and the continuous variables were described as means and standard deviations. Comparisons between the categorical and continuous variables were performed using the Student’s *t*-test and Pearson’s X^2^ test. The tumor markers are expressed as median values and interquartile ranges (IQRs). Overall survival (OS) and disease-free survival (DFS) were analyzed using the Kaplan-Meier analysis with a log-rank test. The prognostic factors of OS were analyzed using a Cox proportional regression analysis, and variables with a *p*-value of < 0.05 were included in a multivariable analysis. To determine the optimal cutoff values that could stratify the survival outcome clearly, the conditional inference tree (C-tree) method, which uses the recursive partitioning of the dependent variables based on the value of the correlation, was used. After obtaining the optimal cutoff values, the values were compared to the upper normal clinical limit of 36 IU/mL for CA 19-9. Multivariate analyses were performed using the Cox proportional hazards model to identify the significant factors that could affect the survival rate. All of the statistical analyses were performed using the SPSS version 25.0 (IBM SPSS Statistics, IBM Corp., Armonk, NY, USA) and R software (version 3.1.2; R Foundation for Statistical Computing, Vienna, Austria). The statistical significance was set at *p* < 0.05.

## 3. Results

### 3.1. Demographics

The mean age was 62 years and with 205 (53.2%) men and 180 (46.8%) women (Table 1). When classified by the T stage, the proportion of patients with the T1 stage was 33.2%, that with the T2 stage was 30.9%, and that with the T3 stage was 33.2%. In terms of the lymph node (LN), the patients were divided into the LN (+) and LN (-) groups. The LN (+) group accounted for 29.6% of patients. The complications were classified according to the Clavien–Dindo grade, and the proportion of grade 3 or higher was 20.5%. The percentage of patients who received adjuvant chemotherapy was 47.3% (Table 1).

### 3.2. Tumor Marker Distribution

The median tumor marker value was 18.6 U/mL (IQR 7.9–67.5) for CA 19-9. The distribution of the preoperative CA19-9 levels is shown in Appendix A. The median values of CA 19-9 were 11.5 U/mL in the T1 stage, 17.0 U/mL in the T2 stage, 37.5 U/mL in the T3 stage, and 153.7 U/mL in the T4 stage (*p* < 0.001). As the T stage increased, the tumor marker levels also showed a significant increase (Appendix A). In the N stage, a pattern of differences similar to the T stage was observed (Appendix A). We additionally analyzed the CA 19-9 level according to pathologic findings. Differences in CA 19-9 level according to tumor size, lymphatic invasion, and venous invasion were not significant. The difference in CA 19-9 level according to perineural invasion was significant. In the perineural invasion negative group, the median value of CA 19-9 was 14.8 U/mL, and in the perineural invasion positive group, it was 40.3 U/mL (*p*-value: 0.033) (Appendix A).

### 3.3. Survival Analysis

In all of the patients, the 5-year overall survival rate (5YOSR) was 61.4%, and the 5-year disease-free survival rate (5YDFSR) was 56.3% (Figure 2a,b). The 5YOSR was analyzed by dividing the patients based on the normal CA 19-9 value of 36 U/mL, which is commonly used in clinical practice (Figure 3). The 5YOSR in the CA 19-9 36 U/mL group was 69.1%, which was higher than the 47.7% in the CA 19-9 > 36 U/mL group (*p* < 0.001). 5YDFSR was analyzed in the same way and showed a similar trend (Figure 4).

### 3.4. The New Optimal Cutoff Survival Value according to the Tumor Marker

The optimal cutoff value for CA 19-9, 46 U/mL was derived using the C-tree method (Figure 5). When the statistical analysis was performed at a level of 46 U/mL of CA 19-9, there was a significant difference between the 5YOSR and 5YDFSR groups (Figure 3 and Figure 4). Table 2 shows the clinicopathological characteristics according to CA 19-9 level of 36 U/mL and 46 U/mL. When the patients were divided into two groups (CA 19-9 level 46 or less than 46 U/mL group and more than 46 U/mL group), there were significant differences in the proportions of the T and N stages, and a significant difference was observed in the proportion of patients who received chemotherapy and radiotherapy (Table 2).

### 3.5. Prognostic Factors for OS

A multivariate analysis was performed to identify the prognostic factors affecting survival. The 5-year survival rates in the group with total bilirubin less than 2.0 mg/dL and the group with total bilirubin 2.0 mg/dL or more were 67.1% and 54.0%, respectively, with a *p*-value of 0.005, which was significant. Therefore, multivariate analysis was performed including total bilirubin. As per the univariate and multivariate analyses, jaundice, histological differentiation, N stage, and adjuvant chemotherapy were significant predictors (Appendix A) (Table 3). When divided by a CA 19-9 level of 36 U/mL, there was a marginal significance with a *p*-value of 0.123 (Appendix A). On the other hand, when divided by a CA 19-9 level of 46 U/mL, it was found to be a statistically significant prognostic factor (*p* = 0.048) (Table 3). Thus, a CA 19-9 level of 46 U/mL is a more effective cutoff than a level of 36 U/mL. The proportion of severe postoperative complications that can affect the prognosis was listed in Table 2. The 5-year survival rates in the less than CD 3a group and the CD 3a or greater group were 62.7% and 54.2%, respectively, with a *p*-value of 0.190, which was not significant. In addition, the ratio of receiving chemotherapy according to the presence or absence of complications also showed no significant difference with a *p*-value of 0.502. Therefore, there was no difference in survival according to the presence or absence of complications (CD 3a or higher).

## 4. Discussion

Tumor marker information can be obtained easily through blood collection [11]. CA 19-9 has been used widely as a factor in determining the diagnosis and prognosis of periampullary cancer, but there is a lack of large-scale studies on AC alone [12,13]. Therefore, we conducted this study using a large cohort and determined a new optimal cutoff value for CA 19-9. Consequently, the value of 46 U/mL was derived using the C-tree method.

Currently, the optimal treatment for AC is pancreaticoduodenectomy (PD). However, there are various options for selecting a treatment strategy for AC. Several studies have suggested surgeries requiring minimal resection for early-stage AC, such as ampullectomy [14]. However, there are some doubtful results with regard to the minimal resection of an AC and the possibility of LN metastases [15]. Moreover, many studies have reported controversial results regarding the extent of lymphadenectomy [16]. Apart from the surgical strategy, the role of adjuvant treatment for AC also remains controversial [17,18,19]. As mentioned above, there are diverse treatment strategies for AC. Moreover, surgical treatment is associated with the following complications: postoperative pancreatic fistulas, postoperative bleeding, and fluid collection [20,21]. Postoperative pancreatic fistulas have been regarded traditionally as the most frequent major serious complication. Therefore, PD may be a treatment option for patients with a high morbidity. So, it is important to determine an appropriate treatment plan, and tumor markers may play a useful role in this.

As a factor that can determine the treatment options for AC, the accuracy of preoperative staging alone is inefficient. Furthermore, the diagnostic accuracy of endoscopic imaging was found to be 67.3% and the diagnostic accuracy of the initial biopsy 67.3%–81.0% [22]. In some cases, lesions that were considered benign according to the biopsy results were found to be malignant after surgery [22]. Surgical treatment, such as PD, may result in significant complications; therefore, there is a need for a method to further evaluate the prognosis before surgery. Since tumor markers can be obtained relatively easily through blood sampling, convenience is guaranteed.

A cohort study of 317 patients demonstrated the efficacy of CA 19-9 as a prognostic factor after a biliary tract cancer resection; however, it was only a study on biliary tract cancer and gallbladder cancer [23,24]. Likewise, for gallbladder cancer alone, another study demonstrated the relationship between the prognosis of gallbladder cancer and the levels of tumor markers, such as CEA and CA 19-9 [25]. Furthermore, a cohort study of 260 patients demonstrated the efficacy of CA 19-9 as a predictor of survival rate and the response to adjuvant chemotherapy, in pancreatic cancer [26]. According to this study, patients with postoperative CA 19-9 levels >90 U/mL did not benefit from adjuvant chemotherapy compared to those with a CA 19-9 of ≤90 U/mL. So far, research has primarily focused on biliary tract cancer rather than AC alone. In a retrospective study that examined the prognostic factors of CA 19-9 in AC, a relatively small number of patients were analyzed. In other words, it has been difficult to apply the major research results to actual clinical practice because the number of patient groups in the study of AC alone has been insufficient [27]. In this study, we conducted large-scale research on AC only.

In addition, from a statistical point of view, existing studies have used mainly current clinical cutoff values without statistical verification. Other studies have analyzed an insufficient number of related patients, thereby lowering the statistical significance. In our study, a new cutoff value was derived using a statistical method, and based on this, a large-scale study was conducted on AC only. Through further research, the new cutoff value can be combined with patient morbidity and added to the pre-operative indication. For example, in an elderly patient with a low morbidity score, if CA 19-9 is less than 46 U/mL, a less aggressive surgical method can be recommended [28,29,30].

Our study had several limitations. This was a single-center retrospective study. In addition, some data were missing, and only those who underwent surgery were included. Patients who received endoscopic treatment or chemotherapy only were excluded. For the larger group, we were unable to combine parameters, such as other images and pathology. In order to increase the reliability of the new cutoff value, it is necessary to apply it to other cohorts. Although it was not possible in this study, Appendix A compares CA 19-9 values by dividing the patients into 3 groups. We believe that this new analysis provides a more comprehensive understanding of the relationship between CA 19-9 levels and survival outcomes in our cohort. Moreover, external validation is planned as an additional separate study. Additionally, there are a certain number of patients who cannot produce CA 19-9 due to a deficiency of the Lewis enzyme, but a lack of information exists in this study. Currently, there are several papers that report on the impact of smoking on CA 19-9. Specifically, one study observed a significant reduction in the geometric mean of CA19-9 levels among current smokers compared to non-smokers, particularly among those with Le/Le (Lewis) genotypes [31]. Our study, however, was limited by inaccurate smoking-related information and a lack of retrospective analysis. In future studies, we will take into account the potential influence of smoking and conduct a comprehensive analysis accordingly. Moreover, both CA 19-9 and total bilirubin were collected just prior to surgery regardless of biliary drainage. Precise analysis according to biliary drainage was excluded from this study. From the perspective of adjuvant radiation therapy, it was excluded from multivariate analysis because it was judged that the effect of adjuvant radiation therapy was unreliable because there were many missing values and the selected criteria were ambiguous.

Nevertheless, our study showed a new cutoff value for the prognosis of AC based on our statistical analysis. Furthermore, we included a large number of cases, much higher than those included in previous studies.

## 5. Conclusions

The prognostic factors for AC were CA 19-9, histologic grade, N stage, and history of adjuvant chemotherapy, and as a tumor marker cutoff value, the CA 19-9 level 46 U/mL rather than 36 U/mL showed statistically significant results. Therefore, tumor markers can be used as important reference data when deciding treatment policies for patients with AC.

## Figures and Tables

**Figure 1 cancers-15-02281-f001:**
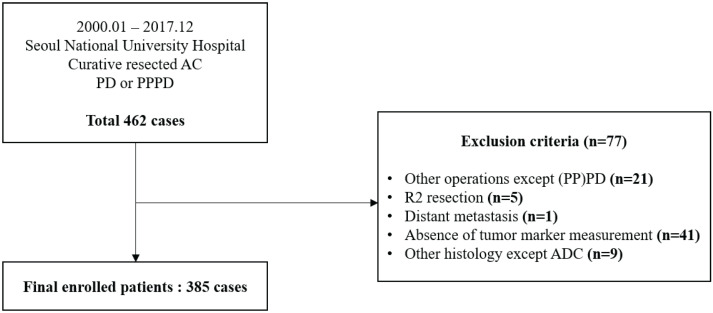
Flow chart of patient selection.

**Figure 2 cancers-15-02281-f002:**
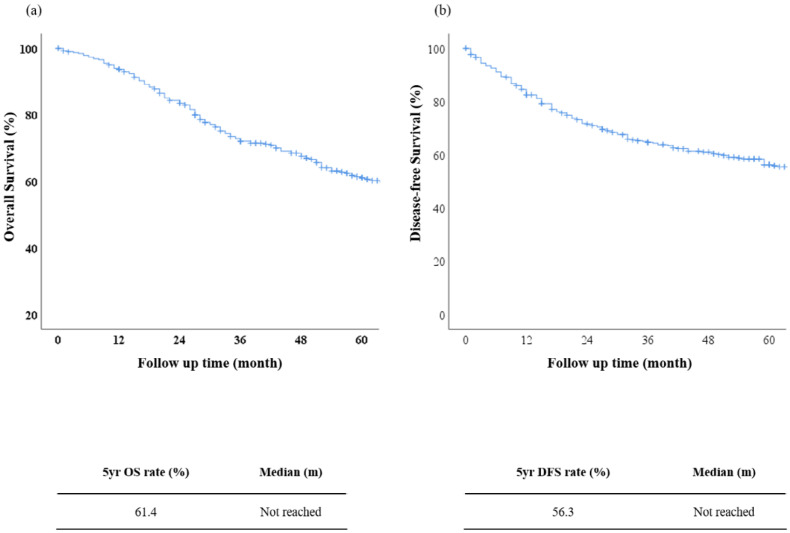
Survival outcomes: (**a**) overall survival; (**b**) disease-free survival. 5-year OS rate; 5-year overall survival rate, 5-year DFS rate; and 5-year disease-free survival rate.

**Figure 3 cancers-15-02281-f003:**
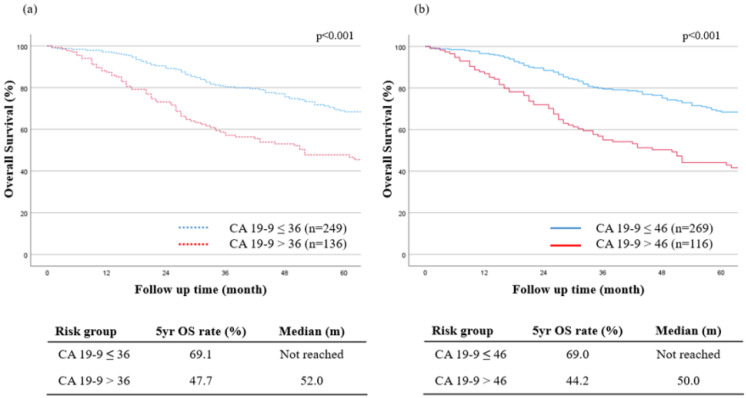
Overall survival according to (**a**) CA 19-9 level 36 U/mL and (**b**) level 46 U/mL.

**Figure 4 cancers-15-02281-f004:**
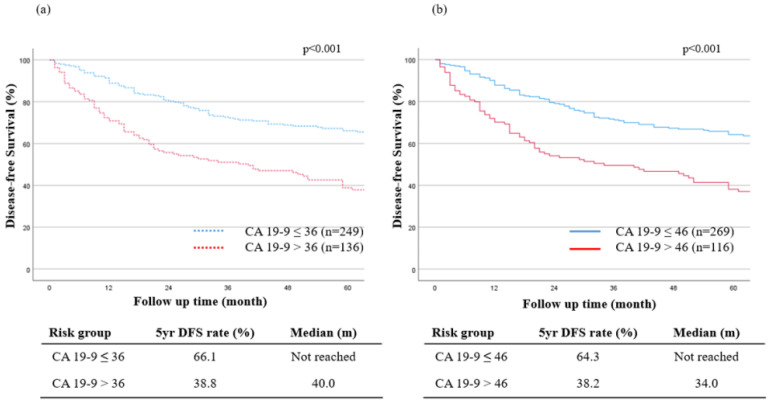
Disease-free survival according to (**a**) CA 19-9 level 36 U/mL and (**b**) level 46 U/mL. 5-year OS rate; 5-year overall survival rate; 5-year DFS rate; and 5-year disease-free survival rate.

**Figure 5 cancers-15-02281-f005:**
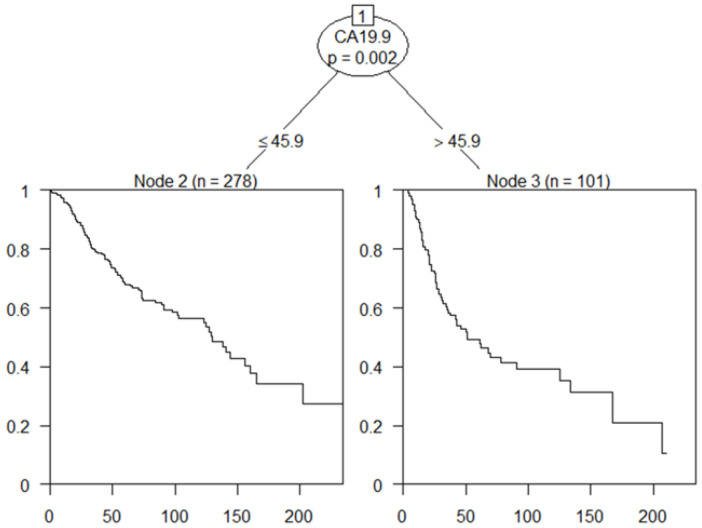
C-tree analysis: the optimal cutoff value of CA 19-9 was 46 U/mL.

**Table 1 cancers-15-02281-t001:** Overall patient demographics and postoperative/pathologic data.

Variables N = 385	
Age, years, mean ± SD	62.45 ± 9.95
Gender, n (%)	
Male	205 (53.2)
Female	180 (46.8)
BMI, kg/m^2^, mean ± SD	23.59 ± 3.28
ASA score, n (%)	
I	125 (32.5)
II	232 (60.3)
III	28 (7.2)
Diabetes mellitus, n (%)	73 (19.0)
CEA, ng/mL, median (IQR)	1.6 (1.1–2.4)
CA 19-9, U/mL, median (IQR)	18.6 (7.85–67.45)
Total bilirubin > 1.3 mg/dL, n (%)	198 (50.1)
Operation time, min, mean ± SD	315.99 ± 79.12
Operation type, n (%)	
Whipple	69 (17.9)
PPPD	316 (82.1)
Operation method, n (%)	
Open	375 (97.4)
Robotic	10 (2.6)
EBL, cc, mean ± SD	398.54 ± 386.95
Differentiation, n (%)	
WD	128 (33.2)
MD & PD	244 (66.8)
Lymphatic invasion (+), n (%)	142 (36.9)
Venous invasion (+), n (%)	31 (8.1)
Perineural invasion (+), n (%)	80 (20.8)
T stage, n (%)	
T1	128 (33.2)
T2	119 (30.9)
T3	128 (33.2)
T4	10 (2.7)
Lymph node (+), n (%)	114 (29.6)
R0 resection, n (%)	383 (99.5)
Hospital stay, day, mean ± SD	20.72 ± 11.47
POPF (+), n (%)	201 (52.2)
Grade B-C POPF (+), n (%)	57 (14.8)
Complication (CD grade ≥3a), n (%)	79 (20.5)
Adjuvant chemotherapy, n (%)	182 (47.3)
Adjuvant radiation therapy, n (%)	158 (41.0)
Recurrence, n (%)	133 (34.5)

BMI: body mass index; ASA score: American Society of Anesthesiologists; CEA: carcinoembryonic antigen; CA 19-9: carbohydrate antigen; PPPD: pylorus preserving pancreaticoduodenectomy; EBL: estimation of blood loss; WD: well-differentiated; MD: moderate-differentiated; PD: poor-differentiated; POPF: postoperative pancreatic fistula; and complication: defined as over Clavien Dindo grade 3a (CD grade 3a).

**Table 2 cancers-15-02281-t002:** Clinicopathologic characteristics according to the CA19-9 levels.

Variables N = 385			CA 19-9 ≤ 36 n = 249	CA 19-9 > 36 n = 136	*p*-Value	CA 19-9 ≤ 46 n = 269	CA 19-9 > 46 n = 116	*p*-Value
Age (mean ± SD, yr)	62.45 ± 9.95		62.19 ± 9.77	62.93 ± 10.32	0.484	62.29 ± 9.73	62.83 ± 10.50	0.628
Gender	Male	205 (53.2%)	135 (54.2%)	70 (51.5%)	0.606	143 (53.2%)	62 (53.4%)	0.958
	Female	180 (46.8%)	114 (45.8%)	66 (48.5%)		126 (46.8%)	54 (46.6%)	
Total bilirubin (mg/dL)	<2.0	215 (55.8%)	162 (65.1%)	53 (39.0%)	<0.001	173 (64.3%)	42 (36.2%)	<0.001
	≥2.0	170 (44.2%)	87 (34.9%)	83 (61.0%)	<0.001	96 (35.7%)	74 (63.8%)	<0.001
Operation type	PPPD	316 (82.1%)	208 (83.5%)	108 (79.4%)	0.313	227 (84.4%)	89 (76.7%)	0.072
	Whipple	69 (17.9%)	41 (16.5%)	28 (20.6%)		42 (15.6%)	27 (23.3%)	
Differentiation	WD	128 (33.2%)	100 (40.2%)	28 (20.6%)	<0.001	102 (37.9%)	26 (22.4%)	0.002
	MD	210 (54.5%)	129 (51.8%)	81 (59.6%)		142 (52.8%)	68 (58.6%)	
	PD	34 (8.8%)	13 (5.2%)	21 (15.4%)		17 (6.3%)	17 (14.7%)	
T stage	T1	128 (33.2%)	106 (42.6%)	22 (16.2%)	<0.001	110 (40.9%)	18 (15.5%)	<0.001
	T2	119 (30.9%)	80 (32.1%)	39 (28.7%)		86 (32.0%)	33 (28.4%)	
	T3	128 (33.2%)	63 (25.3%)	65 (47.8%)		72 (26.8%)	56 (48.3%)	
	T4	10 (2.6%)	0 (0.0%)	10 (7.8%)		1 (0.4%)	9 (7.8%)	
N stage	Negative	271 (70.4%)	197 (79.1%)	74 (54.4%)	<0.001	211 (78.4%)	60 (51.4%)	<0.001
	Positive	114 (29.6%)	52 (20.9%)	62 (45.6%)		58 (21.6%)	56 (48.3%)	
Complication (CD grade ≥3a)	No	306 (79.5%)	200 (80.3%)	106 (77.9%)	0.580	217 (80.7%)	89 (76.7%)	0.379
	Yes	79 (20.5%)	49 (19.7%)	30 (22.1%)		52 (19.3%)	27 (23.3%)	
Chemotherapy	No	203 (52.7%)	148 (59.4%)	55 (40.4%)	<0.001	158 (58.7%)	45 (38.8%)	<0.001
	Yes	182 (47.3%)	101 (40.6%)	81 (59.6%)		111 (41.3%)	71 (61.2%)	
Radiotherapy	No	227 (59.0%)	163 (65.5%)	64 (47.1%)	<0.001	173 (64.3%)	54 (46.6%)	0.001
	Yes	158 (41.0%)	86 (34.5%)	72 (52.9%)		96 (35.7%)	62 (53.4%)	
Recurrence	No	252 (65.5%)	180 (72.3%)	72 (52.9%)	<0.001	192 (71.4%)	60 (51.7%)	<0.001
	Yes	133 (34.5%)	69 (27.7%)	64 (47.1%)		77 (28.6%)	56 (48.3%)	

PPPD: pylorus preserving pancreaticoduodenectomy; WD: well-differentiated; MD: moderate-differentiated; and PD: poor-differentiated.

**Table 3 cancers-15-02281-t003:** Prognostic factors for overall survival (CA 19-9 level cut-off: 46 U/mL).

			Univariate Analysis		Multivariate Analysis	
		Patients (n = 385)	HR (95% CI)	*p*-Value	HR (95% CI)	*p*-Value
Preoperative CEA, ≤5.0/>5.0 (ng/mL)		362/23	1.80 (1.07–3.01)	0.026	1.09 (0.64–1.85)	0.630
Preoperative CA19-9, ≤46/>46 (U/mL)		269/116	2.03 (1.50–2.75)	<0.001	1.37 (1.02–1.88)	0.048
Total bilirubin, <2.0/≥2.0 (mg/dL)		215/170	1.54 (1.14–2.09)	0.005	1.33(0.98–1.82)	0.071
Histologic grade	WD	128	Reference	-	Reference	-
	MD	210	2.29 (1.56–3.35)	<0.001	1.83 (1.24–2.70)	0.002
	PD	34	5.77 (3.45–9.64)	<0.001	4.63 (2.74–7.83)	<0.001
T stage	T1	128	Reference	-	Reference	-
	T2	119	1.87 (1.21–2.90)	0.005	0.99 (0.62–1.59)	0.173
	T3/4	138	3.14 (2.08–4.73)	<0.001	1.35 (0.85–2.15)	0.057
N stage	N0	271	Reference	-	Reference	-
	N+	114	2.82 (2.09–3.79)	<0.001	1.65 (1.19–2.30)	0.003
R status	R0	383	Reference	-	Reference	-
	R1	2	1.16 (0.16–8.30)	0.881	0.67 (0.09–4.90)	0.704
Adjuvant chemotherapy	No	203	Reference	-	Reference	-
	Yes	182	2.84 (2.07–3.90)	<0.001	2.12 (1.49–3.00)	<0.001

WD: well-differentiated; MD: moderate-differentiated; PD: poor-differentiated; HR: hazard ratio; CI: confidence interval; and complication: defined as over Clavien Dindo 3a.

## Data Availability

The data presented in this study are available on request from the corresponding author.

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
