# Peer review of "The Optimal Cutoff Value of Tumor Markers for Prognosis Prediction in Ampullary Cancer"

_cancers, 2023, doi:10.3390/cancers15082281_

Round 1

Reviewer 1 Report

Main Comments:

(1) This manuscript deals with the relationship between the prognosis of ampullary carcinoma and the level of CA 19-9, and with the optimal threshold. The presented evaluation has the limitations of a retrospective study design.

(2) As the data come from a single institution, they reflect the local situation. Further verification of the cutoff value will be necessary.

(3) Can information be provided on the influence of smoking on the CA 19-9 value?

(4) Other potentially confounding variables like cholestasis, inflammation or renal insufficiency should also be considered/discussed.

(5) Please revise the supplementary material (e.g., in the file "Supplement Figure 2", there is a figure legend "Supplement Table 3").

Additional Comments:

(6) The author contributions should be specified.

(7) Conflicts of interest should be disclosed/negated.

Author Response

Response to Reviewer 1

Main Comments:

(1) This manuscript deals with the relationship between the prognosis of ampullary carcinoma and the level of CA 19-9, and with the optimal threshold. The presented evaluation has the limitations of a retrospective study design.

I would like to express my gratitude for your kind remarks regarding my study. I truly appreciate your feedback and valuable insights. I totally agree with your assessment of the limitations inherent in my retrospective study. Your input has been instrumental in helping me recognize areas for improvement and refinement in my research.

Thank you once again for your contribution and for bringing this important point to my attention.

(2) As the data come from a single institution, they reflect the local situation. Further verification of the cutoff value will be necessary.

Thank you for bringing to my attention the positive direction of our work. Your feedback is greatly appreciated and will certainly help to reinforce our efforts.

In line with your suggestion, we plan to conduct additional external validation through cohort data from other hospitals when the results of our journal are recognized. Furthermore, we are actively working on addressing the limitations of our study that you previously highlighted and are eager to explore these areas in future research.

Once again, we really appreciate your valuable input and your continued support.

(3) Can information be provided on the influence of smoking on the CA 19-9 value?

I would like to express my gratitude for your nice comments.

Currently, there exist several papers that report on the impact of smoking on CA 19-9. Specifically, one study observed a significant reduction in the geometric mean of CA19-9 levels among current smokers compared to non-smokers, particularly among those with Le/Le (Lewis) genotypes. Our study, however, was limited by inaccurate smoking-related information and a lack of retrospective analysis.

In future studies, we will take into account the potential influence of smoking and conduct a comprehensive analysis accordingly.

This information has been added to the discussion section of the text.

Your kind words and support are much appreciated.

Thank you again for taking the time to share your comments.

Reference>

Kawai S, Suzuki K, Nishio K, Ishida Y, Okada R, Goto Y, Naito M, Wakai K, Ito Y, Hamajima N. Smoking and serum CA19-9 levels according to Lewis and secretor genotypes. Int J Cancer. 2008 Dec 15;123(12):2880-4. doi: 10.1002/ijc.23907. PMID: 18803289.

(4) Other potentially confounding variables like cholestasis, inflammation or renal insufficiency should also be considered/discussed.

I would like to express my gratitude for your insightful comment regarding our study. Your feedback is highly valued and has certainly contributed to our understanding of the factors that may affect our findings.

As you mentioned, there are several factors that can influence the results, including cholestasis, inflammation, and renal insufficiency. To exclude the effect of cholestasis, we measured CA19-9 after bilirubin level was decreased when ERBD was inserted immediately before surgery. We also added and analyzed the bilirubin level 2.0 mg/dL criterion, but did not find any significant results. In the case of inflammation, we targeted patients with average preoperative inflammation markers such as WBC, neutrophil counts, and CRP, and found no significant difference among the patient groups. Regarding renal insufficiency, we were unable to analyze it in this study due to the inaccuracy of old medical records. However, we acknowledge the importance of this factor and will consider it in future studies.

We greatly appreciate your feedback and suggestions, and we agree that multi-center studies including external validation may help to provide a more comprehensive understanding of these issues. Once again, thank you for bringing this to our attention and for your continued support.

(5) Please revise the supplementary material (e.g., in the file "Supplement Figure 2", there is a figure legend "Supplement Table 3").

Please allow me to express our sincere apologies for any confusion caused by the material presented in our study. We appreciate your feedback and are grateful for the opportunity to clarify and improve our work.

As per your suggestion, we have modified the Supplementary material accordingly, and we hope that the revised version provides greater clarity and accuracy for our readers.

Once again, we would like to extend our deepest apologies for any inconvenience caused, and we thank you for bringing this matter to our attention.

Additional Comments:

(6) The author contributions should be specified.

Thank you for your valuable input regarding our work. Your feedback has been instrumental in helping us refine and improve our study.

As per your suggestion, we will amend the relevant section as follows:

[HK contributed to the conception and design of this study.

SL, HK, and YH analyzed and interpreted the data.

SL, HK wrote the manuscript.

All authors participated in patient recruitment and collected and assembled data. 

All authors have proofread the manuscript.]

We appreciate your keen attention to detail and your commitment to ensuring the accuracy and rigor of our research. Your contribution is greatly appreciated.

Thank you once again for your insightful comment and for your continued support.

(7) Conflicts of interest should be disclosed/negated.

Thank you for notifying me regarding the contents. I appreciate your prompt communication and attention to detail.

To ensure we are both on the same page, I would like to confirm that the contents are as follows:

[Disclosures

The authors declare no conflicts of interest.]

Please let me know if you have any further questions or concerns. Thank you again for your time and cooperation.

Reviewer 2 Report

The paper, The optimal cutoff value of tumor markers for prognosis prediction in ampullary cancer, proposes to alter the cutoff of the CA19-9 marker level from 36U/mL to 46U/ml as a prognostic indicator, to spare some patients from overly aggressive treatment.  In their cohort of 385, some 20 patients would be shifted into the low expression category. This has little impact on OS curve for low expressers, while producing a small impact on the high expression set as suggested by C-tree analysis and shown by multivariate analysis of preoperative CA levels.

The paper could present some results in a better fashion.  It took me a while to see the difference between Fig 4a,b and S Fig 3. If possible can these be layered on top of each other, so the difference can just seen in one figure instead of flipping between two sets of very similar graphs?  Or, could you graph the OS and DFS curves of the 20 altered subjects onto the graph in S Fig 3?  Just consider making some change in presentation.  Even adding grid lines with labels for the key quantifications to the graphs might help. 

Can you comment on the 20 patients shifted by the new cutoff?  They seem to have OS and DFS similar to the low expressing set.  As suggested above, this could be shown on the OS and DFS graphs.  And/or with comments on general outcome in the text

Author Response

Response to Reviewer  2 Comments

The paper, The optimal cutoff value of tumor markers for prognosis prediction in ampullary cancer, proposes to alter the cutoff of the CA19-9 marker level from 36U/mL to 46U/ml as a prognostic indicator, to spare some patients from overly aggressive treatment.  In their cohort of 385, some 20 patients would be shifted into the low expression category. This has little impact on OS curve for low expressers, while producing a small impact on the high expression set as suggested by C-tree analysis and shown by multivariate analysis of preoperative CA levels.

I would like to express my gratitude for your insightful comment regarding our study. Your feedback is greatly appreciated and has contributed to our understanding of the potential implications of the CA 19-9 cutoff value.

As you mentioned, changing the CA 19-9 cutoff value from 36 to 46 may have little influence on the analysis for the 20 patients newly classified in the cohort of this study. However, in order to secure a sufficient sample size, we recognize the importance of collecting and validating multi-institutional data.

Therefore, we plan to expand our study to include data from multiple institutions in order to increase the statistical power of our analysis and improve the generalizability of our findings.

Once again, please accept my sincere appreciation for your valuable input and for your continued support.

The paper could present some results in a better fashion.  It took me a while to see the difference between Fig 4a,b and S Fig 3. If possible can these be layered on top of each other, so the difference can just seen in one figure instead of flipping between two sets of very similar graphs?  Or, could you graph the OS and DFS curves of the 20 altered subjects onto the graph in S Fig 3?  Just consider making some change in presentation.  Even adding grid lines with labels for the key quantifications to the graphs might help.

Thank you for your valuable feedback regarding our study. Your comments have been extremely helpful in improving the visual representation of our data.

As per your suggestion, we will be making the following changes:

Fig 4. and S Fig 3. will be merged into Fig 4. for improved clarity and ease of interpretation.

Survival analysis according to CA 19-9 levels of 36 U/mL and 46 U/mL will be presented side by side for more efficient visualization. When overlapping the two graphs, the visual effect is rather reduced due to the subtle difference, so the two graphs can be viewed side by side.

A grid line will be added to improve readability and help distinguish between the two graphs.

We appreciate your keen attention to detail and your commitment to ensuring the accuracy and clarity of our work. Your contribution has been invaluable in improving the overall quality of our study.

Thank you once again for your insightful comment and for your continued support.

Can you comment on the 20 patients shifted by the new cutoff?  They seem to have OS and DFS similar to the low expressing set.  As suggested above, this could be shown on the OS and DFS graphs.  And/or with comments on general outcome in the text

Thank you for your positive feedback regarding our work. We appreciate your interest in our study and your support for our research.

As per your request, we have updated the S Figure 3. in order to present a new survival analysis. This analysis divides the CA 19-9 level into three groups, based on the following cutoff points: CA 19-9 level 36, 36 < level 46, and more than 46.

We believe that this new analysis provides a more comprehensive understanding of the relationship between CA 19-9 levels and survival outcomes in our cohort. As shown in the graph, the group with CA 19-9 level above 36 U/mL and below 46 U/mL actually has a better prognosis than the group above 46 U/mL. We hope that you find this updated figure informative and helpful. The graph above can be seen in detail in the attached file.

Once again, thank you for your kind words and your ongoing interest in our work. 

Round 2

Reviewer 2 Report

The authors have addressed my remarks adequately.